# Improving oral health in people with severe mental illness (SMI): A systematic review

**Alexandra Macnamara** [1]*, **Masuma Pervin Mishu**[2], **Mehreen Riaz Faisal**[2], **Mohammed Islam**[3], **Emily Peckham**[2]

1 The University of York and Hull York Medical School, Castle Hill Hospital, York, United Kingdom,
2 Department of Health Sciences, The University of York, York, United Kingdom, 3 Hull University Teaching Hospitals NHS Trust, Hull, United Kingdom

* Alexandra.Macnamara@hyms.ac.uk

## Abstract

### Background

Those with severe mental illness (SMI) are at greater risk of having poor oral health, which can have an impact on daily activities such as eating, socialising and working. There is currently a lack of evidence to suggest which oral health interventions are effective for improving oral health outcomes for people with SMI.

### Aims

This systematic review aims to examine the effectiveness of oral health interventions in improving oral health outcomes for those with SMI.

### Methods

The review protocol was registered with PROSPERO (ID CRD42020187663). Medline, EMBASE, PsycINFO, AMED, HMIC, CINAHL, Scopus and the Cochrane Library were searched for studies, along with conference proceedings and grey literature sources. Titles and abstracts were dual screened by two reviewers. Two reviewers also independently performed full text screening, data extraction and risk of bias assessments. Due to heterogeneity between studies, a narrative synthesis was undertaken.

### Results

In total, 1462 abstracts from the database search and three abstracts from grey literature sources were identified. Following screening, 12 studies were included in the review. Five broad categories of intervention were identified: dental education, motivational interviewing, dental checklist, dietary change and incentives. Despite statistically significant changes in plaque indices and oral health behaviours as a result of interventions using dental education, motivational interviewing and incentives, it is unclear if these changes are clinically significant.

**Data Availability Statement:** All relevant data are within the manuscript and its Supporting Information files.

**Funding:** The authors received no specific funding for this work.

**Competing interests:** The authors have declared that no competing interests exist.

## Conclusion

Although some positive results in this review demonstrate that dental education shows promise as an intervention for those with SMI, the quality of evidence was graded as very low to moderate quality. Further research is in this area is required to provide more conclusive evidence.

## Introduction

Severe mental illness (SMI) is an umbrella term, which aims to encompass those who experience serious functional impairment as a result of their mental illness. Within the UK, the definition of SMI includes those with schizophrenia, bipolar disorder and other non-organic psychotic conditions [1]. The prevalence of SMI in England is 0.96%, although it is also thought to be a significant problem in a global context [2, 3]. There is a need to consider physical health in this population, as those with SMI have a higher prevalence of cardiovascular disease, liver disease, respiratory disease, diabetes and cancer [4–10]. The effects of these health inequalities are significant, as those with SMI die on average 15 to 20 years earlier than the general population [5].

Oral health problems are also more prevalent in those with SMI [11–13]. One systematic review found that this population have 2.8 times the odds of being having no teeth compared with the general population, and, on average, had five more decayed, missing or filled teeth compared to people without SMI [12]. Risk factors for oral health diseases include high sugar intake, smoking and alcohol consumption, with these behavioural risk factors more prevalent in those with mental illness [14–16]. Those with SMI may also have specific risk factors for poor oral health, such as dry mouth as a side effect of medication, and may face barriers, such as lack of motivation, negative attitudes of dental practitioners, anxiety around visiting the dentist and costs of dental treatments [12].

Poor oral health can have a significant impact on quality of life affecting social life, self-esteem and social interactions [17]. Poor oral health also impacts on health functioning, for example problems with eating, which can have subsequent effects on nutrition [18]. In addition, it is associated with other physical health conditions, such as diabetes [19], pneumonia and cardiovascular disease [20, 21].

Current guidance from NHS England and the British Society for Disability and Oral Health (BSDH) has reflected on the need for improved oral health outcomes in those with mental health problems and has highlighted the importance of the common risk factor approach [22–24]. In the common risk factor approach, it is recognised that sugar consumption, smoking and poor oral hygiene are shared risk factors for different chronic health conditions [24]. However, despite this guidance, there is a lack of evidence of effective approaches to oral health in this population.

One previous Cochrane review considered the role of oral health education in improving oral health in those with SMI [25]. Overall, despite some statistically significant improvements in patients' dental plaque index, the authors concluded that there was no clinically significant benefit from oral health education [25]. However, oral health education is only one form of a wide range of possible interventions that may improve oral health. As the previous Cochrane review did not cover the full range of potential oral health interventions, this review intends to build on the previous systematic review by considering a larger range of interventions that may be used to promote good oral health in this population.

The aim of this review is to examine the range and effectiveness of oral health interventions on any measure of oral health in those with SMI, to understand which oral health interventions are effective in improving oral health in those with severe mental illness.

## Methods

This systematic review has been reported according to the Preferred Reporting Items for Systematic Review and Meta-Analysis (PRISMA) guidelines (S1 File) [26].

A protocol for the systematic review was developed and registered with the PROSPERO database prior to the review being conducted (ID CRD42020187663). The protocol is available from: https://www.crd.york.ac.uk/prospero/display_record.php?RecordID=187663

### Eligibility criteria

We used a specialized framework, called PICO, to formulate the research question and eligibility criteria, and develop our search strategy [27]. The PICO framework encompasses the patient population or problem, intervention, comparison and outcome [27].

**Participants.** Eligible participants were defined as adults (aged over 18) diagnosed with SMI. For the purpose of this review, we considered SMI to be people with schizophrenia, bipolar disorder or other psychotic disorders (including other disorders with psychotic features, such as delusional disorders). Studies were also considered for inclusion if at least 75% of the participants fit the SMI criteria, if the authors defined their participants as having SMI or if separate results were available for those with SMI. Studies with participants with a dual psychiatric diagnosis that included SMI, as defined above, were considered eligible.

**Intervention.** Any intervention that was implemented with the aim of improving oral health was considered eligible.

**Comparator.** Any comparator group was considered eligible.

**Outcome.** Any outcome that was either a direct clinical measure of changes in oral health or a proxy measure of oral health, such as toothbrushing habits, were considered eligible.

**Inclusion criteria.** All studies with an interventional study design were considered eligible.

No date restrictions were imposed, although only English language studies were included due to lack of resources available for translation.

**Exclusion criteria.** Studies were excluded if the population did not meet the criteria of SMI (as defined above) or if the participants were children. Studies that did not include an oral health intervention were excluded, as were studies that did not have an outcome relevant to oral health. Non-interventional studies, including qualitative studies, and non-English studies were also excluded.

### Data sources

A search strategy was developed based on the pre-defined eligibility criteria (S2 File). This search strategy was used to perform systematic searches of PubMed (1946–2021), Embase (1974–2021), the Allied and Complementary Medicine Database (AMED) (1985–2021), the Health Management Information Consortium (HMIC) (1979–2021), the Cochrane Library (no date range), Cumulative Index to Nursing and Allied Health Complete (CINAHL Complete) (inception to 2021), Scopus (inception to 2021) and PsycINFO (1967–2021) during June 2021. The reference lists of all studies retained for full text screening were searched for eligible studies. To identify possible sources of grey literature, searches of OpenGrey, an open access database of grey literature, was performed [28]. Conference abstracts prior to May 2020 were also searched from two international Psychiatry conferences.

## Study selection

Two reviewers (MM, MF) independently screened all titles and abstracts against the defined eligibility criteria. If at any point it was unclear whether the study met the eligibility criteria, it was retained for full text screening. Rayyan [29] was used for title and abstract screening.

Full text screening was performed by four reviewers (AM, MM, MF and MI), with each study independently screened by two reviewers. There were no disagreements in relation to study inclusion. Full text screening was completed using Covidence software [30].

## Data extraction

Data extraction was undertaken using Covidence software [30]. After piloting the template on two studies, the tool was amended to ensure all information was captured. Data extraction included data on study funding, study methodology, participant details, study setting, intervention and control description and results on study outcomes.

Data extraction for each study was performed independently by two reviewers during July and August 2021.

## Risk of bias assessment

To assess the risk of bias in RCTs, version two of the Cochrane risk-of-bias tool for randomised trials (RoB 2) was used at the outcome level [31]. Where studies presented multiple outcomes, the results for clinical measures were used for the risk of bias assessment. For cluster RCTs, RoB 2 for cluster-randomised studies was used [32].The risk of bias in non-randomised studies was assessed using the Risk of Bias in Non-Randomised Studies–of Interventions (ROBINS—I) tool and, where applicable, the Cochrane guidance for applying the ROBINS-I tool to uncontrolled before and after studies was applied [33]. All studies were included in the data synthesis regardless of the risk of bias outcome. The risk of bias assessment for each study was undertaken independently by two reviewers.

The quality of evidence was assessed using the Grading of Recommendations Assessment, Development and Evaluation (GRADE) approach [34].

## Synthesis approach

A comparison of the studies found significant variation in relation to the clinical context and methodology of the included studies, therefore, a narrative synthesis was used for data synthesis within this review as a meta-analysis would have been inappropriate.

The studies were grouped by outcome and intervention type and a descriptive narrative of the results was produced.

# Results

## Study selection

The database search yielded 2,073 results, with 1,462 studies found in total after removal of duplicates. A further three abstracts were identified through searches of grey literature sources.

During the database search, two protocols were identified for studies that would potentially meet the eligibility criteria for the systematic review. The corresponding authors were contacted to ask if they would be able to provide any unpublished data. One author confirmed that the study was still in progress and no response was received from the author of the second study.

Conference abstracts were also reviewed from two large international Psychiatry conferences. As no potential studies were identified following this handsearching, it was deemed

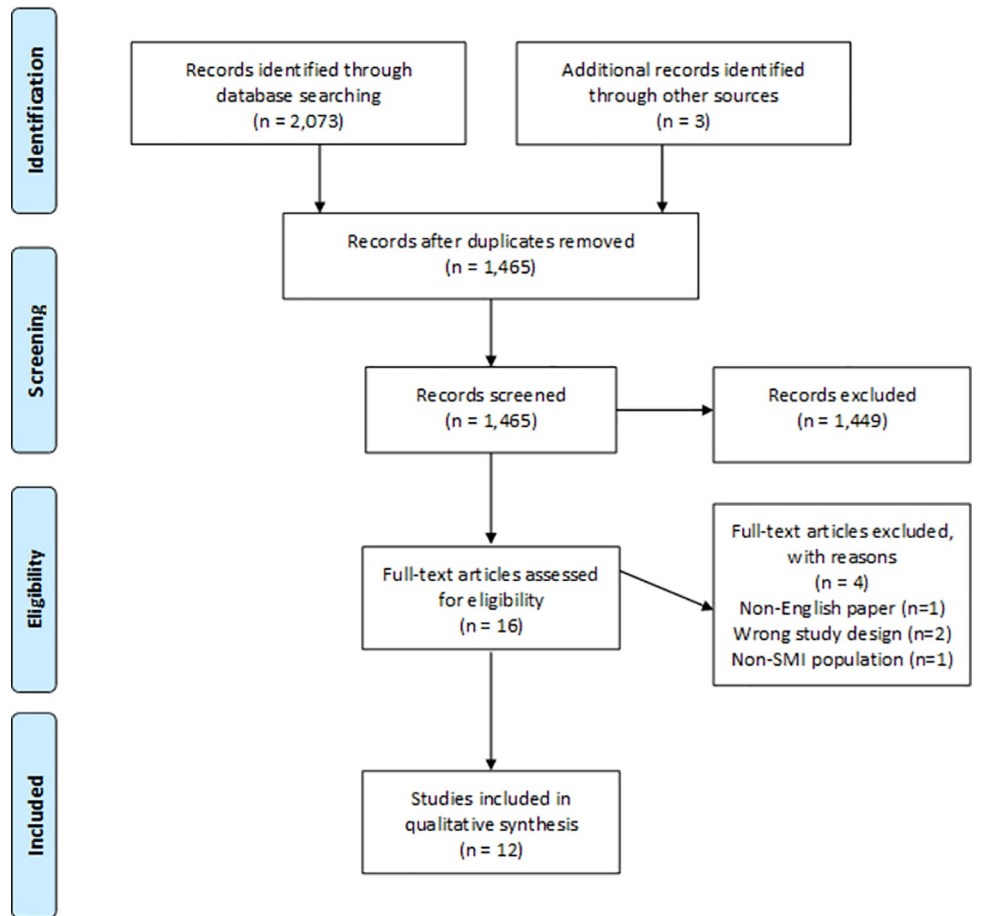

**Fig 1. PRISMA flowchart.**

unlikely that further searching would yield relevant results and no further conference abstracts were searched.

In total, 1,462 abstracts from the database search and three abstracts from grey literature were screened. From all sources searched, the full texts of 16 studies were screened for eligibility, and of these 16 studies, 12 were included in the systematic review [35–46]. Reasons for exclusion included having a non-interventional study design (n = 1) [47], having a population which did not meet the criteria for SMI (n = 1), being a commentary on an included study (n = 1) [48] and not being published in English (n = 1) [49] (Fig 1) [26].

## Characteristics of included studies

The 12 included studies varied in terms of date, country, study design, population, intervention and outcome measures. Seven of the included studies were RCTs [35, 36, 39, 43–45], with two using a cluster RCT design [42, 44]. Two studies used a quasi-experimental study design [37, 46] and the remaining three studies were uncontrolled before and after studies [38, 40, 41]. A summary of the study characteristics is provided in Table 1.

**Participants.** The sample size varied significantly between studies, from 10 participants [41] to over 1682 participants (the exact number of participants in this study was unknown) [42]. Four studies focused purely on schizophrenic patients [37, 39, 41, 43] and six had participants with a variety of diagnoses [35, 36, 38, 40, 44, 45].

**Table 1. Characteristics of included studies.**

| Author, Year, Country | Study Design | Participants | Intervention | Control | Outcomes (related to oral health) |
|---|---|---|---|---|---|
| Adams et al 2018 [42] | Cluster RCT | n = 35 teams; unknown number of participants (>1682) | Dental checklist for care coordinators | No checklist | • Number visiting a dentist within 12 months |
| | | | | | • Registration with dentist |
| | | | | | • Routine check-up within last 12 months |
| UK | | **Diagnosis**: Suspected psychosis | | | • Owning a toothbrush |
| | | | | | • Brushing twice a day |
| | | | | | • Non-routine visit to a dentist in the last year |
| Agarwal et al 2019 [39] | RCT | n = 111 | Oral health education | Standard oral care advice | • Oral health knowledge and attitudes |
| | | | | | • Oral hygiene practices |
| India | | **Diagnosis:** Schizophrenia | | | • Oral health (gingival index, debris index, calculus index and oral hygiene index) |
| Almomani et al 2009 [35] | RCT | n = 60 | Motivational interviewing | Oral health education session only | • Oral health (modified Quigley-Hein Plaque Index) |
| USA | | **Diagnosis:** Schizophrenia, bipolar affective disorder and depression | | | • Oral health knowledge |
| | | | | | • Self-regulation for toothbrushing (Treatment Self-Regulation Questionnaire) |
| Almomani et al 2006 [36] | RCT | n = 50 | Oral health education | Provision of mechanical toothbrush only | • Patients' oral health (modified Quigley-Hein Plaque Index) |
| USA | | **Diagnosis:** Schizophrenia, Bipolar Disorder, Depression | | | |
| de Mey et al 2016 [38] | Before and after study | n = 27 | Oral health education for nurses and patients | N/A | • Nursing staff oral health knowledge |
| The Netherlands | | **Diagnosis:** Schizophrenia/ schizoaffective disorder, bipolar disorder, anxiety, autism, borderline personality disorder and amnesia | | | • Oral health (plaque index and bleeding index) |
| Jean et al 2020 [43] | RCT | n = 60 | Mangosteen fruit plus non-surgical therapy | Non-surgical therapy only | • Oral health (plaque index, bleeding index, periodontal pocket depth and clinical attachment level) |
| India | | **Diagnosis:** Chronic schizophrenia | | | |
| Gottfried and Verdicchio 1974 [40] | Before and after study | n = 50 | Incentivisation (rewards provided for hygiene related behaviours) | N/A | • Proportion of participants brushing their teeth |
| USA | | **Diagnosis:** Schizophrenia, "psychotic with mental deficiency", personality disorder and organic brain disorder | | | |
| Klinge 1979 [41] | Before and after study | n = 10 | Oral health education | N/A | • Frequency of unhealthy oral health behaviours |
| USA | | **Diagnosis:** Schizophrenia | | | |
| Kuo et al 2020 [44] | Cluster RCT | n = 68 | Composite intervention including group and individual education and incentives | Standard nursing care | • Oral health knowledge, attitudes and behaviour |
| Taiwan | | **Diagnosis:** Schizophrenia, bipolar disorder, major depressive disorder and any organic mental illness | | | • Oral health (plaque index) |

*(Continued)*

**Table 1.** (Continued)

| Author, Year, Country | Study Design | Participants | Intervention | Control | Outcomes (related to oral health) |
|---|---|---|---|---|---|
| Mun et al 2014 [45] | RCT | n = 88 | Different forms of oral health education | Different forms of oral health education | • Oral health (Patient Hygiene Performance plaque index, acid production and Lactobacillus) |
| Korea | | **Diagnosis**: Schizophrenia, schizoaffective disorder, bipolar disorder, depression, organic mental disorder | | | • Measures of dry mouth (salivary flow, oral dryness) |
| Nurbaya et al 2020 [37] | Quasi-experimental study | n = 104 | Oral health education | No education | • Oral health knowledge and attitudes |
| Indonesia | | **Diagnosis:** Schizophrenia | | | • Oral health (plaque index) |
| Singhal et al 2021 [46] | 2x2 Quasi-experimental study | n = 103 | Home care instructions and battery-operated toothbrush | Control groups included no home care instructions and manual toothbrush | • Oral health (gingival index and plaque index) |
| USA | | **Diagnosis:** Axis I psychiatric disorder | | | |

One study described participants as being diagnosed with an Axis I psychiatric disorder [46] and one did not include details of participants' diagnoses, with participants classified as "people with or at risk of serious mental illness" [42]. It was decided to include this study as this population group are likely to face similar symptoms to those with SMI, and thus may face similar barriers to accessing dental care [50]. This study also included a small proportion of participants under 18, however as the majority of participants were over 20 years old, the study was included.

The age of participants ranged from 15 to 83 years. Three of the studies had all male participants [40, 41, 44] and four had a larger proportion of men than women [37, 39, 42, 46]. One study did not report demographic information of the participants [43].

**Setting.** Studies were conducted in the United States [35, 36, 40, 41, 46], India [39, 43], Indonesia [37], the UK [42], Korea [45], Taiwan [44] and the Netherlands [38]. Clinical settings for the studies included community-based care [35, 36, 42], inpatient populations [40, 41, 44] and outpatients [37, 39, 46]. One study recruited patients from both inpatient and outpatient settings [38] and one study included inpatients and those attending a daytime programme [45]. One study did not provide details of the clinical setting [43].

**Intervention.** The interventions were grouped into five main categories: dental education, motivational interviewing, dental checklist, incentives and dietary changes.

Seven of the interventions incorporated oral health education, although other components were also implemented, including provision of toothbrushes, reminder systems for brushing and staff education [36–39, 41, 44]. Six of these included a control group, receiving usual nursing care [37, 44], standard oral health advice [39], different educational materials [45], no education or manual toothbrush [46] or provision of a mechanical toothbrush only [36]. Two studies also delivered oral health education to mental health staff [38, 41].

One study focussed on motivational interviewing (MI) alongside dental education, and compared this intervention to a control group that received the oral health education only [35].

One study trialled the implementation of a dental checklist [42]. Alongside the checklist, dental awareness training was provided, however this was in the context of general training related to the study [42].

Two studies included the provision of incentives following completion of tasks or attendance at educational sessions, one study using this as the only intervention [40] and another as part of a composite intervention [44].

The impact of dietary changes on oral health was considered in one study, in which mango-steen fruit was used as an adjunct to non-surgical therapy [43].

**Outcomes.** The study outcomes encompassed clinical measures of oral health, measures of xerostomia (dry mouth), measures of oral health knowledge, beliefs or behaviours, or a combination of different outcomes.

The clinical measures of oral health involved a number of different oral health indices and measures of bacterial activity. One study performed ultrasonic scaling on all participants' teeth after baseline examination, in order to then compare follow up scores with a baseline of zero [39].

Measures of oral health behaviours included the proportion of participants brushing their teeth as advised [39, 40, 42, 44], the proportion undertaking unhealthy brushing behaviours [41] and the number accessing dental care [42, 44]. In the study evaluating dental checklists, the checklist was used as both the intervention and the outcome measurement tool [42].

Some of the studies assessed the change in oral health knowledge in patients [35, 37, 44] or nursing staff [38].

Four studies also sought to assess motivations and attitudes towards oral health following an oral health intervention using questionnaires [35, 37, 39, 44], including the Dental Coping Beliefs Scale (DCBS) [39] and the Treatment Self-Regulation Questionnaire (TSRQ) [35].

## Risk of bias

Four RCTs [35, 36, 39, 45] were categorised as "some concerns" in relation to overall risk of bias, and one RCT was found to be at high risk of bias overall [43] as displayed in Fig 2 [51].

The two studies using a cluster RCT study design were both categorised as "some concerns" in relation to the overall risk of bias (Fig 3).

The ROBINS-I tool was used for risk of bias assessment in the non-randomised studies (Fig 4) [51, 52].

Fig 2. Risk of bias in RCTs.

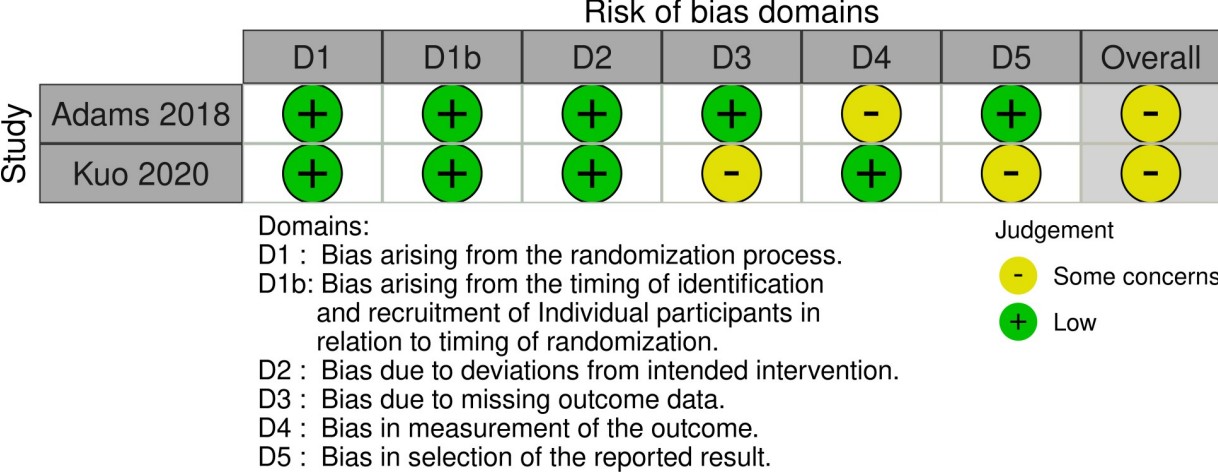

Fig 3. Risk of bias in cluster RCTs.

As displayed in Fig 4, all studies were classified as being at serious risk of bias except one study which was found to be at critical risk of bias [40].

## Publication bias

To minimise the risk of publication bias specialist databases [53], conference proceedings and grey literature sources were searched. However it is possible that there may be language bias due to exclusion of non-English studies [54].

### Risk of bias domains

| Study | D1 | D2 | D3 | D4 | D5 | D6 | D7 | Overall |
|---|---|---|---|---|---|---|---|---|
| De Mey 2015 | X | + | + | + | + | + | - | X |
| Gottfried 1974 | X | X | X | - | - | X | X | ! |
| Klinge 1979 | X | - | - | - | - | - | - | X |
| Nurbaya 2020 | X | + | + | + | + | - | - | X |
| Singhal 2021 | X | + | + | + | - | X | - | X |

Domains:
D1: Bias due to confounding.
D2: Bias due to selection of participants.
D3: Bias in classification of interventions.
D4: Bias due to deviations from intended interventions.
D5: Bias due to missing data.
D6: Bias in measurement of outcomes.
D7: Bias in selection of the reported result.

Judgement
! Critical
X Serious
- Moderate
+ Low

Fig 4. Risk of bias in non-randomised studies.

Due to the significant heterogeneity in outcome measures across the included studies, a funnel plot could not be generated to assess for publication bias [55].

## Data synthesis

To allow for data synthesis, studies were grouped by outcome measures and intervention type. The outcomes were classified as clinical measures of oral health, measures of dry mouth, oral health knowledge and attitudes and oral health behaviours. The interventions were grouped into dental education, motivational interviewing, dental checklist and incentivisation. The impact of different intervention types on each kind of outcome measure was considered. A summary of the outcomes relating to oral health is shown in Table 2 and a summary of the key findings related to oral health behaviours attitudes is shown in Table 3.

## Impact of interventions on clinical measures of oral health

**The impact of dental education on clinical measures of oral health.** One study found a statistically significant difference between the intervention and control groups in the debris index scores (2.28±0.93 vs 2.98±1.3, p = 0.02) and oral hygiene index (5.19±2.2 vs 6.27±2.5, p = 0.04) [39]. However, the difference between the intervention and control group in the mean scores for gingival index (1.45±0.46 vs 1.47±0.51) and calculus index (2.90±1.5 vs 3.28 ±1.5) at follow up were not statistically significant [39].

In relation to plaque scores, one study found a greater improvement of 0.8 in the intervention group (from 3.0±0.2 to 2.2±0.2), compared with a change of 0.6 in the control group (from 3.3±0.2 to 2.7±0.2) [36]. The between group differences were found to be statistically significant, with a p value of 0.026 [36].

Different forms of oral health education were compared in one RCT [45]. In this study, group A received toothbrushing training plus video education and a brochure, group B received the video education and brochure, and group C received the brochure only. At 12 weeks, there was found to be a statistically significant difference between the Patient Hygiene Performance Index (2.07 vs 1.72 vs 2.32, p = 0.036), although the lowest final score was seen in group B [45]. However, each group was found to have a statistically significant decrease in plaque after each session (p<0.0001) [45]. The study found no significant group differences in acid production (68.55 vs 70.08 vs 66.55) or measures of lactobacillus (1.95 vs 1.65 vs 1.73) [45].

Two quasi-experimental studies also evaluated the effects of oral health education on clinical outcomes [37, 46]. One study compared a group receiving toothbrushing education with a control group and found a statistically significant difference in plaque index scores at two weeks (1.31 vs 2.78, p = 0.000) [37]. The other non-randomised study used a 2x2 between subjects study design, comparing four groups [46]. Group A received oral health education and an electric toothbrush, Group B received an electric toothbrush only, Group C received education and a manual toothbrush and Group D received a manual toothbrush only [46]. When comparing the groups that received education with those that did not, there was no statistically significant effect on plaque index or gingival index, however there was found to be a statistically significant impact of the type of toothbrush on the gingival index (p<0.05), but not plaque index [46].

Clinical measures of oral health were also used in one uncontrolled before and after study. An improvement was seen in the six surface and two surface plaque index measures, with a change from median scores of 73 to 42 (p<0.01) and 56 to 23 (p<0.01), respectively [38]. The change in median bleeding index scores using six surface measurements also showed a statistically significant improvement from 40 to 32 (p<0.05), however the two surface bleeding index score did not show a significant improvement, with a change from 25 to 23 (p>0.05) [38].

**Table 2. Summary of the intervention effects on clinical measures of oral health.**

| Study details, intervention and numbers in final analysis | Outcome measure | Result |
|---|---|---|
| Agarwal et al 2019 [39] | **Gingival index** (final mean score and SD) | **1.45 (0.46) vs 1.47 (0.51)** (p = 0.80) |
| **Education (n = 41) vs control (n = 45)** | **Debris index** (final mean score and SD) | **2.28 (0.93) vs 2.98 (1.30)** (p = 0.02) |
| | **Calculus index** (final mean score and SD) | **2.90 (1.50) vs 3.28 (1.50)** (p = 0.18) |
| | **Oral hygiene index** (final mean score and SD) | **5.19 (2.2) vs 6.27 (2.5)** (p = 0.04) |
| Almomani et al 2009 [35] | **Plaque index** (final mean score and SD) | **1.9 (0.7) vs 2.58 (0.9)** (p<0.01) |
| **MI and education (n = 27) vs control (n = 29)** | | |
| Almomani et al 2006 [36] | **Plaque index** (final mean score and SD) | **2.2 (0.2) vs 2.7 (0.2)** (p = 0.026). |
| **Education (n = 20) vs control (n = 22)** | | |
| de Mey et al 2016 [38] | **Plaque index—6 surface** (pre and post median score) | **73 to 42** (p<0.016) |
| **Education: pre and post test scores (n = 24)** | **Plaque index– 2 surface** (pre and post median score) | **56 to 23** (p<0.01) |
| | **Bleeding index– 6 surface** (pre and post median score) | **40 to 32** (p<0.05) |
| | **Bleeding index– 2 surface** (pre and post median score) | **25 to 23** (p = 0.29) |
| Jean et al 2020 [43] | **Plaque index** (final mean score) | **0.8 vs 0.9** (p = 0.70) |
| | **Gingival index** (final mean score) | **0.56 vs 1.05** (p = 0.05) |
| **Dietary change (n = 30) vs control (n = 30)** | **Periodontal probing depth** (final mean score) | **4.50 vs 4.74** (p = 0.38) |
| | **Clinical attachment level** (final mean score) | **4.12 vs 4.43** (p = 0.26) |
| Kuo et al 2020 [44] | **Plaque index** (final mean score and SD) | **42.6 (12.1) vs 61.8 (11.6)** (p<0.001) |
| **Education and incentives (n = 27) vs control (n = 31)** | | |
| Mun et al 2014 [45] | **Patient Hygiene Performance Index** (final mean score and SD) | **2.07 (1.06) vs 1.72 (0.84) vs 2.32 (1.00)** (p = 0.036) |
| **Education intervention (n = 23) vs video education and brochure (n = 22) vs brochure only (n = 28)** | **Cariview® (acid production)** (final result and SD) | **68.55 (7.59) vs 70.08 (6.64) vs 66.55 (8.97)** (p≥0.05) |
| | **Dentocult LB® (lactobacillus)** (final result and SD) | **1.95 (1.15) vs 1.65 (0.75) vs 1.73 (1.16)** (p≥0.05) |
| Nurbaya et al 2020 [37] | **Plaque index** (final mean score and SD) | **1.31 (0.65) vs 2.78 (0.68)** (p = 0.000) |
| **Education (n = 52) vs control (n = 52)** | | |
| Singhal et al 2021 [46] | **Plaque index** (mean change) | **-0.38 vs -0.23** (p>0.05) |
| **Education (n = 44) vs no education (n = 43)** | **Gingival index** (mean change) | **-0.38 vs -0.48** (p>0.05) |

**The impact of motivational interviewing (MI) with dental education on clinical measures of oral health.** One study evaluated the effect of MI in combination with dental education.

When comparing baseline results to those at week eight, the MI group showed a greater improvement in the plaque index score from 3.6 (±0.6) to 1.9 (±0.7), compared to a change from 3.3 (±0.8) to 2.5 (±0.9) in the control group. The final results showed a significant difference in the plaque scores between the MI and control group (p<0.01) [35].

**The impact of dental education with incentives on clinical measures of oral health.** One cluster RCT conducted by Kuo et al trialled a composite intervention which included individual and group education, along with reminders and incentives in the form of tokens

**Table 3. A summary of the intervention effects on oral health attitudes, behaviours, and knowledge.**

| Study | Outcome Measure | Result |
|---|---|---|
| Adams et al 2018 [42]<br><br>**Checklist (n = 123) vs control (n = 172)** | **Registering with dentist** (log RR and 95% CI, Yes vs No) | **0.13 (-0.20, 0.45)** (p = 0.44) |
| | **Having routine check-up in the last year** (log RR and 95% CI, Routine check-up vs visit to fix a problem) | **-0.20 (-0.51, 0.10)** (p = 0.18) |
| | **Owning a toothbrush** (log RR and 95% CI, Yes vs No) | **-0.01 (-2.14, 2.12)** (p = 0.99) |
| | **Cleaning teeth twice a day** (log OR and 95% CI, cleaning twice daily vs < twice daily) | **-0.09 (-0.57, 0.37)** (p = 0.68) |
| | **Requiring urgent dental treatment** (log RR and 95% CI, Yes vs No) | **0.49 (-0.11, 1.10)** (p = 0.11) |
| Agarwal et al 2019 [39]<br><br>**Education (n = 41) vs control (n = 45)** | **Brushing teeth twice daily** (%) | **23.2% vs 7.3%** (p = 0.04) |
| | **Having a toothbrush** (%) | **91.1% vs 72.7%** (p = 0.04) |
| | **Changing toothbrush every 3 months** (%) | **73.2% vs 47.3%** (p = 0.01) |
| | **Perception of their oral health as good** (%) | **69.6% vs 89.1%** (p = 0.01) |
| | **Internal locus (Dental Coping Belief Scale)** (Mean final score and SD) | **20.12 (1.81) vs 8.41 (3.0)** (p<0.00) |
| | **External locus (Dental Coping Belief Scale)** (Mean final score and SD) | **10.26 (3.1) vs 11.80 (2.6)** (p<0.01) |
| | **Self-efficacy (Dental Coping Belief Scale)** (Mean final score and SD) | **13.75 (1.7) vs 12.07 (1.9)** (p<0.00) |
| | **Oral health beliefs (Dental Coping Belief Scale)** (Mean final score and SD) | **7.62 (2.0) vs 9.5 (2.0)** (p<0.00) |
| Almomani et al 2009 [35]<br><br><br><br>**MI and education vs control** | **Oral health knowledge** (mean final score and SD) | **32.9 (1.7) vs 27.5 (4.3)** (p<0.01) |
| | **Introjected regulation (Self-regulation Scores)** | **6.1 (1.3) vs 5.0 (2.0)** (p<0.01) |
| | **External Regulation (Self-regulation Scores)** | **3.6 (2.1) vs 3.4 (2.2)** (Not significant) |
| | **Autonomous regulation (Self-Regulation scores)** | **4.0 (2.3) vs 3.3 (2.0)** (Not significant) |
| de Mey et al 2016 [38]<br><br>**Education: pre and post test (n = 27)** | **Nursing oral health knowledge** (Mean pre and post test scores and SD) | **11.33 (3.63) to 14.30 (2.22)** (p<0.001) |
| Gottfried and Verdicchio 1974 [40]<br><br>**Incentives: pre and post test (n = 50)** | **Proportion brushing teeth** (%) | Baseline: **8%** |
| | | Day one **88%** (p<0.01) |
| | | Throughout programme **98%** |
| Klinge 1979 [41]<br><br>**Education: pre and post test (n = 10)** | **Frequency of unhealthy oral health behaviours** (significance only) | Change in need for brushing reminders (p<0.001), refusal to brush (p<0.01) and using dentrifice but no brush (p<0.05). No significant change in other parameters. |
| Kuo 2020 [44]<br><br>**Education and incentives (n = 27) vs control (n = 31)** | **Oral health knowledge** (final mean score) | **8.5 vs 4.7** (p<0.001) |
| | **Oral health attitudes** (final mean score) | **58.6 vs 45.8** (p<0.001) |
| | **Oral health behaviours** (final mean score) | **8.1 vs 3.8** (p<0.001) |
| Nurbaya 2020 [37]<br><br>**Education (n = 52) vs control (n = 52)** | **Oral health knowledge** (final mean score and SD) | **14.79 (0.96) vs 10.62 (1.66)** (p = 0.000) |
| | **Oral health attitudes** (final mean score and SD) | **55.83 (4.21) vs 52.19 (2.91)** (p = 0.000) |

which could be spent on food or toiletries. The results for the plaque index at 12 weeks found a statistically significant difference between the intervention and control group (42.6 vs 61.8. p<0.001) [44].

**The impact of dietary changes on clinical measures of oral health.** One study considered the effects of mangosteen fruit as an adjunct to deep scaling and root planning [43]. After three months, the results found a statistically significant difference in gingival scores between the intervention and control group (0.56 vs 1.05, p = 0.05). However, there was no statistically significant difference between the two groups for plaque index (0.8 vs 0.9, p = 0.70), periodontal probing depth (4.50 vs 4.74, p = 0.38) and clinical attachment level (4.12 vs 4.43, p = 0.26) [43].

A summary of the effects of the interventions on clinical measures of oral health is displayed in Table 2.

## Impact of interventions on measures of dry mouth

Only one study had outcomes relating to measures of dry mouth, including saliva production and subjective oral dryness [45]. Comparing three groups with different forms of oral health education, the results found that none of the groups had statistically significant changes in salivary production or subjective oral dryness scores over time, however results for oral dryness scores were significant for all participants combined (p = 0.015) [45].

## Impact of interventions on oral health knowledge and attitudes

**The impact of dental education on oral health knowledge and attitudes.** The effects of oral health education on oral health beliefs as measured by the DCBS showed a positive impact. Internal locus scores for the intervention group showed a significant improvement when compared to the control group (20.12±1.81 vs 8.41±3.0, p<0.001), as did the external locus scores (10.26±3.1 vs 11.80±2.6, p<0.01) [39]. There was also improvement in the education group in comparison to the control group in self-efficacy (13.7±1.7 vs 12.07±1.7, p<0.001) and oral health beliefs (7.6±2.0 vs 9.5±2.0, p<0.001) [39].

One study considered the change in oral health knowledge of nursing staff, with a change in mean score from 11.33 to 14.30 out of 16 (p<0.001) [38].

Oral health knowledge and attitudes were measured in the quasi-experimental study conducted by Nurbaya et al. At two weeks, there were statistically significant difference between the education and control group for both oral health knowledge (14.79 vs 10.62, p = 0.000) and oral health attitudes (55.83 vs 52.19, p = 0.000) [37].

**The impact of dental education with incentives on oral health knowledge and attitudes.** One study using a composite intervention, including aspects of education and incentives, in a cluster RCT looked at the impacts on oral health knowledge and attitudes [44]. This study used a questionnaire adapted from a tool by the Taiwan Health Promotion school and included aspects of knowledge around oral diseases and oral health, as well as attitudes about dental care [44]. The results show statistically significant differences at 12 weeks between the intervention and control group for both oral health knowledge (8.5 vs 4.7, p<0.001) and oral health attitudes (58.6 vs 45.8, p<0.001) [44].

**The impact of MI with dental education on oral health knowledge and attitudes.** Results from the TSRQ at week eight showed a statistically significant main effect of intervention in favour of the MI group when compared to the control group (6.1±1.3 vs 5.0±2.0, p = 0.027) for introjected regulation, whereas for the results for external regulation (3.6±2.1 vs 3.4±2.2) and autonomous regulation (4.0±2.3 vs 3.3±2.0), there were no significant changes [35].

## Impact of interventions on oral health behaviours

**The impact of dental education on oral health behaviours.**   The impact of oral health education on oral health practices was measured in one RCT. Questionnaire responses showed statistically significant differences between the number of participants in the education and control groups in the frequency of brushing twice daily, changing toothbrush every three months, use of aid for cleaning teeth (toothbrush), using sulcular brushing technique, using a soft brush, use of aid for cleaning tongue and massaging gums after brushing [39]. The results for rinsing after every meal did not show significant differences between the two groups [39].

Another study considered the impact of oral health education on the frequency of unhealthy oral health behaviours, with statistically significant reductions in the frequency of patients' need for reminders to brush, refusal to brush and using dentifrice with no brush. The remaining variables showed no statistically significant improvement [41].

**The impact of dental education with incentives on oral health behaviours.**   The composite intervention trialled in one cluster RCT found a significant difference between oral health behaviours in the intervention and control group at 12 weeks (8.1 vs 3.8, p<0.001) [44].

**The impact of a dental checklist on oral health behaviours.**   In one cluster RCT, the impact of a dental checklist was evaluated. There were found to be no statistically significant differences between the intervention and control group in registering with a dentist, attending for a check-up, owning a toothbrush, brushing twice daily and requiring urgent dental treatment [42].

**The impact of incentives on oral health behaviours.**   With the use of incentives, the rate of toothbrushing improved from a 8% to 88% on day one, with the researchers reporting that the percentage levelled off at 98% [40].

Table 3 provides a summary of the effects on oral health behaviours and knowledge.

## Quality of evidence in this systematic review

The main outcomes that were considered for quality assessment were ones related to clinical parameters as these were considered the most relevant outcomes to inform guidelines in relation to oral health (Table 4).

Overall, the quality of evidence for all outcome domains was classified as moderate, low or very low. The main quality issue, which was considered across all studies, was imprecision. Although the optimal information size was not calculated for the included studies, the sample sizes in the majority of studies were small. A funnel plot was not generated due to differences in the way intervention effects were measured across studies, which meant that publication bias could not be considered in the quality assessment. However, other factors were considered in the quality of evidence assessment, such as risk of bias arising from other aspects than those included in the GRADE approach.

## Discussion

The synthesis of results found statistically significant changes in clinical measures of oral health as a result of interventions using dental education, motivational interviewing, a composite intervention incorporating elements of education and incentives and dietary change. There were also positive changes in oral health behaviours as a result of education and incentives, although the results for this were inconsistent with a combination of statistically significant and insignificant results. In addition, oral health education appeared to lead to some improvement in plaque and debris indices, with the quality of evidence ranging between very low and moderate quality. However, across the different study designs, there was no convincing evidence that education led to improvements in oral health indices and oral health

**Table 4. Quality assessment of the evidence using the GRADE approach.**

| Outcome | Number of studies | Study design | Risk of bias | Inconsistency | Indirectness | Imprecision | Other factors | Overall |
|---|---|---|---|---|---|---|---|---|
| **Comparison: Oral health education vs no education** | | | | | | | | |
| **Plaque/ debris index** [36, 37, 39] | 3 | RCTs (2) and quasi-experimental study (1) | No serious risk of bias | No serious inconsistency | Serious[a] | Serious[b] | Not serious | Moderate ⊕⊕⊕⊖ |
| **Gingival/ bleeding index** [39] | 1 | RCT | No serious risk of bias | No serious inconsistency | Serious[a] | Serious[c] | Not serious | Low ⊕⊕⊖⊖ |
| **Comparison: Toothbrushing training with educational materials vs other forms of education** | | | | | | | | |
| **Plaque/ debris index** [45] | 1 | RCT | No serious risk of bias | No serious inconsistency | No serious indirectness | Serious[c] | Not serious | Moderate ⊕⊕⊕⊖ |
| **Gingival/ bleeding index** | 0 | - | - | - | - | - | - | No data |
| **Comparison: Oral health education vs no education (incorporating different toothbrushes)** | | | | | | | | |
| **Plaque/ debris index** [46] | 1 | 2x2 quasi-experimental study | Serious risk of bias | No serious inconsistency | Serious[a] | Serious[b] | Not serious | Very low ⊕⊖⊖⊖ |
| **Gingival/ bleeding index** [46] | 1 | 2x2 quasi-experimental study | Serious risk of bias | No serious inconsistency | Serious[a] | Serious[b] | Not serious | Very low ⊕⊖⊖⊖ |
| **Comparison: Oral health education with no control** | | | | | | | | |
| **Plaque/ debris index** [38] | 1 | Uncontrolled before and after study | Serious risk of bias | No serious inconsistency | No serious indirectness | Serious[b] | Not serious | Low ⊕⊕⊖⊖ |
| **Gingival/ bleeding index** [38] | 1 | Uncontrolled before and after study | Serious risk of bias | No serious inconsistency | No serious indirectness | Serious[b] | Not serious | Low ⊕⊕⊖⊖ |
| **Comparison: Composite intervention (education and incentives) vs standard care** | | | | | | | | |
| **Plaque/ debris index** [44] | 1 | Cluster RCT | No serious risk of bias | No serious inconsistency | Serious[a] | Serious[c] | Not serious | Low ⊕⊕⊖⊖ |
| **Gingival/ bleeding index** | 0 | - | - | - | - | - | - | No data |
| **Comparison: MI and oral health education vs oral health education** | | | | | | | | |
| **Plaque/ debris index** [35] | 1 | RCT | No serious risk of bias | No serious inconsistency | Not serious | Serious[b] | Serious[d] | Low ⊕⊕⊖⊖ |
| **Gingival/ bleeding index** | 0 | RCT | - | - | - | - | - | No data |
| **Comparison: Dental checklist vs no checklist** | | | | | | | | |
| **Plaque/ debris index** | 0 | - | - | - | - | - | - | No data |
| **Gingival/ bleeding index** | 0 | - | - | - | - | - | - | No data |
| **Comparison: Incentives with no control** | | | | | | | | |
| **Plaque/ debris index** | 0 | - | - | - | - | - | - | No data |
| **Gingival/ bleeding index** | 0 | - | - | - | - | - | - | No data |
| **Comparison: Dietary change with non-surgical treatment vs non-surgical treatment only** | | | | | | | | |
| **Plaque/ debris index** [43] | 1 | RCT | Serious risk of bias | No serious inconsistency | Serious[a] | Serious[b] | Not serious | Very low ⊕⊖⊖⊖ |
| **Gingival/ bleeding index** | 0 | - | - | - | - | - | - | No data |

GRADE certainty ratings

Very low: The true effect is probably markedly different from the estimated effect.

Low: The true effect might be markedly different from the estimated effect.

Moderate: The authors believe that the true effect is probably close to the estimated effect.

High: The authors have a lot of confidence that the true effect is similar to the estimated effect.

[a] May not be generalisable to all with SMI

[b] Lack of information around precision of results

[c] Some imprecision around results and not large sample size

[d] Corporate sponsor

[e] Wide confidence intervals for all domains.

behaviours due to a combination of statistically significant and non-significant results. Furthermore, some of the study authors acknowledged that, despite improvements, the oral health of participants remained in a relatively unhealthy state [38, 39].

Interestingly, in relation to perceptions around oral health, the oral health education group in one study had worse perceptions of their oral health in comparison to the control group, despite the clinical measurements suggesting that oral health was poorer in the control group [39]. However, research also suggests that lack of self-awareness around oral health may contribute to the neglect of oral care, meaning that an increased awareness of oral health could lead to increased motivation and improved oral care [56].

Health education and advice have shown promise in promoting healthy lifestyle behaviours in previous research [57], and one trial specifically found beneficial effects in physical health promotion in those with mental illness [58]. However, a Cochrane review of oral health education for those with SMI, which included three of the studies within this review [35, 36, 42], concluded that there was no evidence that oral health education resulted in "clinically meaningful outcomes" [25]. Although the results of this systematic review appear to be in-line with the previous Cochrane review, other research has been conducted examining the effectiveness of oral health education in the general population, which found positive oral health outcomes as a result of dental education [59]. However, defining interventions as oral health education may be an oversimplification of such interventions, although oral health education can be considered to encompass advice and training [25], models such as the behaviour change wheel consider different and distinct forms of interventions within this umbrella term [60]. Furthermore, people with SMI may require an educational intervention tailored to their specific needs.

MI is considered a useful strategy in health promotion, with evidence suggesting that it can be effective in encouraging a range of lifestyle changes [61]. Some have considered whether this person-centred approach may be beneficial in helping translate knowledge into action by bringing about motivation to change [62, 63]. Although individual studies have shown promise [64], a systematic review evaluating the effectiveness of MI in oral health found no conclusive evidence that MI is effective at improving oral health [63]. Echoing the results for individual studies, in this review one study was identified that included MI as an intervention with the results demonstrating statistically significant differences on plaque index and oral health knowledge [35]. MI is beginning to be increasingly used in those with mental illness [65] and promising results for the use of MI adaptations for smoking cessation in those with SMI [66] suggest that an intervention combining MI with other approaches may be appropriate for this population.

One study evaluated the use of a dental checklist completed by care co-ordinators, however the results suggested that the checklist did not have any effect on oral health behaviours [42]. This checklist, based on BSDH guidelines, could be considered as an intervention that focuses on environmental restructuring [42]. One of the most common uses of checklists in healthcare is to ensure patient safety, for example the use of the World Health Organisation Surgical Safety Checklist [67]. Other studies that have demonstrated the utility of checklists include those designed to improve documentation of ward rounds [68], improve completion of relevant physical health and screening checks [69] and improve physical health screening in psychiatric wards [70]. These studies aimed to use the checklist as a prompt for completing specific actions, such as completion of physical observations. The checklist intervention in this review may have acted as a reminder, however it was not necessarily designed to lead to specific actions being completed by the care-coordinator [42]. This checklist therefore acts as a different kind of reminder, as the person completing the checklist in this study was not directly responsible for completion of the checklist items. It could also be the case that dental checklists implemented in an inpatient setting may demonstrate different results.

Incentives have been used in a variety of ways to encourage behaviour change, predominantly in the form of financial incentives [71, 72]. From a theoretical basis, it is thought that incentives may help individuals in their transition through stages of change from "pre-contemplation" or "contemplation" to "preparation" or "action" [73, 74]. Although incentives are still used in some settings, the evidence base for the effectiveness of incentives is mixed, with one systematic review suggesting that they may be effective for leading to a variety of health-related behaviour changes [75], whereas a review of incentives purely for weight loss found no significant effect on weight change [76]. In this review, two studies provided results on the use of incentives [40, 44]. One study considered the effects of incentives with education as part of a composite intervention, encompasses other aspects such as group and individual education [44]. In the other study that considered incentives, the incentives offered within this study were tobacco and "candy" [40], which can also be considered as an unethical intervention due to the potential harm associated with the incentives [77]. Incentives can be considered as positive or negative, however this may be an oversimplification [72]. Although the incentives within this study are seemingly positive, with the provision of a desirable substance on completion of the tasks, it could also be seen as a negative incentive for those who do not perform the tasks, as these substances had previously been freely available before the research. The nature of the research and "rewards" themselves mean that it is difficult to compare the results of this study with other research trialling incentives, such as the provision of vouchers for smoking cessation in pregnancy [78].

The final intervention type that was considered in this review was dietary changes in the form of mangosteen fruit acting as an adjunctive treatment [43]. Other dietary interventions have been trialled to evaluate the impact of oral health, such as the impact of specific fruits including kiwi and grapefruit, however the results demonstrated varying significance of the fruits on oral health [79, 80]. A more comprehensive dietary intervention was trialled in a small pilot study, which found improvements in some, but not all oral hygiene and periodontal indices [81]. Although this was not trialled within the SMI population, promoting a healthy diet could have an important role in oral health, particularly as part of the common risk factor approach [24].

## Strengths and limitations of the review

This is the first systematic review to consider a range of different intervention types for improving oral health in those with SMI. Pre-defined methods were adhered to and our search strategy aimed to identify studies from a range of sources. However, this review did not consider studies that were not published in English, due to lack of resources available for translation.

In relation to the risk of bias assessments for the three non-randomised studies included, the ROBINS-I tool was used. However, the non-randomised studies included were all uncontrolled before and after studies, meaning that not all domains and signalling questions included in the ROBINS-I guidance were relevant to this study design. Although Cochrane provide guidance on additional and alternative questions that may be considered in order to make this tool more applicable to these studies [33], using a tool that relies on the comparison of a control group for some of the signalling questions, this may mean that the risk of bias assessment may not be appropriate for these studies.

Some of the intervention and outcome categories were only addressed by one study, some of which were found to be at a high risk of bias, meaning that any conclusions drawn from this limited evidence should be done so with caution.

The eligibility criteria were designed to be as broad as possible in order to include a variety of interventions and encompass different aspects of oral health. However, with more broad

inclusion criteria, there is a risk of these criteria appearing ambiguous, which may increase the risk of subjectivity when considering studies for inclusion [82]. Although some elements of the inclusion criteria such as population were described in detail in order to reduce this risk, other aspects such as intervention or outcome could be considered ambiguous. The study design aimed to minimise this risk through having two independent reviewers conduct both title and abstract, as well as full text screening.

## Implications for practice

The included studies have shown that both oral health education, in association with dental training and provision of toothbrushes, and MI, may have an impact on oral health. These limited data may therefore have implications regarding improvement in oral hygiene. However, although these results demonstrated a statistically significant change of plaque score favouring the intervention group, it is unclear if these results are clinically important.

Incentivisation may also have a positive effect on oral health and toothbrushing behaviours, however it would also require careful consideration of how incentives could be used in this context to avoid disadvantaging specific groups.

Previous authors have recommended that in the absence of convincing evidence for particular interventions, the BSDH guidelines should be followed [25, 42]. One important aspect of these guidelines includes the requirement for a patient-centred approach to oral health promotion [22]. Those with SMI are not a homogenous group, and the approach to improving oral health should consider individual health needs, such as diagnosis, severity of illness, living arrangements and physical health [83–85]. The clinical setting is also likely to impact on the way interventions and approaches can be implemented [22, 86], so an important element of any future guidance should consider how the range of approaches to oral health promotion may be applied to different settings and patient groups, and how it may work in collaboration with existing strategies for physical health promotion.

Another perspective on improving physical health in those with SMI, including oral health, is the idea that all professionals should take an active role in health promotion and monitoring [22, 85, 86]. Recommendations from the BSDH guidance include suggestions on the completion of oral health assessments and care plans, educating healthcare professionals to enable them to address oral health needs and multi-disciplinary collaboration [22].

The evidence from this review suggests that oral health education could have a role as an intervention for those with SMI, however the current evidence is limited. This review has also begun to explore some of the alternative interventions that have also been trialled in this population. It is not recommended that clinical practice should be changed based on this review, rather the results should assist clinicians, policy makers and public health professionals in beginning to ask questions about how oral health can be improved in this population group, how existing guidelines can be more uniformly implemented and how the included studies relate to existing guidelines and practice.

## Implications for further research

Further research should be completed before any of the oral health interventions that have been discussed are recommended. It is likely that a stronger evidence base will be provided by RCTs with larger sample sizes, which may further explore if some interventions or approaches are more effective in different clinical settings or for different patient groups.

Another important aspect to be considered in any future research is the longer terms effects of oral health interventions, with longer follow up periods, to consider whether the intervention has any longer lasting effects.

In addition, it is recommended that the impact of more holistic oral health care plans and interventions that aim to improve access to dental services should be investigated.

Finally, further research needs to consider which approach would be the most appropriate for this study population, as, in general, there was little consideration of the theoretical underpinning for the use of selected interventions in the included studies. More careful reasoning and justification for studied interventions may allow for more specifically tailored approaches for oral health promotion in those with SMI. An important aspect of this would be to include engagement with patient groups. Although some important qualitative research has been conducted in this area [87, 88], further work could be done to understand other aspects of interventions in relation to acceptability and how well approaches would address existing barriers to accessing dental care and maintaining good oral health.

## Conclusion

Those with SMI are at greater risk of having poor oral health, due to the many barriers this group face in maintaining good oral health. This systematic review aimed to evaluate the effectiveness of oral health interventions in those with SMI. A total of twelve studies were included in this review, which encompassed four broad categories of intervention.

Although some statistically significant results were seen in improving dental indices and toothbrushing behaviours, particularly in relation to oral health education, there was no convincing evidence that these interventions led to clinically significant changes in oral health. The strongest evidence was found in relation to the effects of oral health education on plaque index, which was assessed as being of moderate quality, whereas the overall quality for the remaining categories of evidence were assessed as being either low or very low.

Recommendations have been made for further research opportunities in order to form stronger conclusions about the effectiveness of oral health interventions in this population. Finally, in relation to clinical practice, the BSDH guidelines can still be considered a useful framework to promote oral health in those with mental illness. From a public health perspective, the common risk factor approach should be considered a key element in addressing oral health inequalities in those with SMI.

## Supporting information

**S1 File. PRISMA checklist.**
(DOCX)

**S2 File. Search strategy for EMBASE.**
(DOCX)

## Author Contributions

**Conceptualization:** Masuma Pervin Mishu, Emily Peckham.

**Data curation:** Alexandra Macnamara.

**Formal analysis:** Alexandra Macnamara.

**Investigation:** Alexandra Macnamara, Masuma Pervin Mishu, Mehreen Riaz Faisal, Mohammed Islam.

**Methodology:** Alexandra Macnamara, Masuma Pervin Mishu, Mehreen Riaz Faisal, Mohammed Islam, Emily Peckham.

**Supervision:** Masuma Pervin Mishu, Emily Peckham.

**Writing – original draft:** Alexandra Macnamara.

**Writing – review & editing:** Alexandra Macnamara, Masuma Pervin Mishu, Mehreen Riaz Faisal, Mohammed Islam, Emily Peckham.

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
