## [Decision Letter · Decision Letter 0]

27 Apr 2021

PONE-D-21-11259

Improving Oral Health in People with Severe Mental Illness (SMI): A Systematic Review

PLOS ONE

Dear Dr. Macnamara,

Thank you for submitting your manuscript to PLOS ONE. After careful consideration, we feel that it has merit but does not fully meet PLOS ONE’s publication criteria as it currently stands. Therefore, we invite you to submit a revised version of the manuscript that addresses the points raised during the review process.

The reviewers have identified significant concerns on your submission. Please carefully consider all the reviewers' comments and address them fully if you plan to resubmit.

We look forward to receiving your revised manuscript.

Kind regards,

Ratilal Lalloo

Academic Editor

PLOS ONE

Journal Requirements:

Reviewers' comments:

**Comments to the Author**

1. Is the manuscript technically sound, and do the data support the conclusions?

Reviewer #1: Partly

Reviewer #2: Partly

Reviewer #3: Yes

2. Has the statistical analysis been performed appropriately and rigorously? 

Reviewer #1: N/A

Reviewer #2: N/A

Reviewer #3: N/A

3. Have the authors made all data underlying the findings in their manuscript fully available?

Reviewer #1: Yes

Reviewer #2: Yes

Reviewer #3: Yes

4. Is the manuscript presented in an intelligible fashion and written in standard English?

Reviewer #1: Yes

Reviewer #2: No

Reviewer #3: Yes

5. Review Comments to the Author

Reviewer #1: Thank you very much for the opportunity to review the paper entitled “Improving Oral Health in People with Severe Mental Illness (SMI): A Systematic Review.” In this work, the authors try to summarize the results of studies that focused on interventions to improve the oral health of individuals with SMI such as schizophrenia and bipolar disorders. The topic is interesting since few reviews have been conducted on the topic and authors tried to be comprehensive in terms of interventions and oral health outcomes. That being said, there are some concerns regarding the methodology and preparation of the report that need to be addressed before the paper can be considered for publication.

Introduction

The introduction is well written and informative for readers who are not familiar with interdisciplinary topics such as the current paper. Some previous studies and reviews were mentioned and the gap in knowledge was described.

Methods

It is suggested that the participant, intervention, comparison, and outcome (PICO) framework be used to formulate the research focused question. In that way, readers will have a more clear idea of what to expect from the paper.

Eligibility Criteria

Page5, L 88. What authors mean by adults? What is the age range?

Page 5, L 90. It is suggested that authors describe more in detail the other psychotic disorder since readers in the field of dentistry may not be familiar with them.

Authors need to clearly define inclusion and exclusion criteria under separate subheadings so it would be easier for readers to follow.

Page 5 L 92. What do authors mean by dual diagnosis exactly? SMI and another psychological disorder?

Data Sources

Authors need to mention the exact month for their systemic search in 2020 for each database.

Study Selection

Why 90% of the screening of titles and abstracts were done by one author? It’s not a common approach in conducting systematic reviews and usually 2 authors screen all the records. Moreover, does 100% agreement between the two authors, when piloting for 10% of the records, refer to those papers that were excluded, included, or both? If it only refers to excluded papers, it can be a source of bias.

Possible Missed Papers in the Review

Only seven papers were included in this review which consequently prevents authors from conducting meta-analysis due to heterogeneity. By a quick search only in PubMed, the following papers were found that possibly could have been included in this review.

1. Singhal V, Heuer AJ, York J, Gill KJ. The Effects of Oral Health Instruction, and the Use of a Battery-Operated Toothbrush on Oral Health of Persons with Serious Mental Illness: A Quasi-Experimental Study. Community mental health journal. 2020:1-8.

2. Peteuil A, Rat C, Moussa-Badran S, Carpentier M, Pelletier J-F, Denis F. A therapeutic educational program in oral health for persons with schizophrenia: a qualitative feasibility study. International journal of dentistry. 2018;2018.

3. Kuo M-W, Yeh S-H, Chang H-M, Teng P-R. Effectiveness of oral health promotion program for persons with severe mental illness: a cluster randomized controlled study. BMC oral health. 2020;20:1-9.

4. Denis F, Millot I, Abello N, Carpentier M, Peteuil A, Soudry-Faure A. Study protocol: a cluster randomized controlled trial to assess the effectiveness of a therapeutic educational program in oral health for persons with schizophrenia. International journal of mental health systems. 2016;10:1-10.

5. Jones HF, Adams CE, Clifton A, et al. An oral health intervention for people with serious mental illness (Three Shires Early Intervention Dental Trial): study protocol for a randomised controlled trial. Trials. 2013;14:1-8.

Possibly by conducting a complete search in other databases more papers will be found that have the potential to be included. It’s suggested that authors conduct the literature search again and include any missed papers and revise the Result and Discussion section accordingly.

Results

Page 8 L 162. Only one reference in the bracket was mentioned which seems to be by mistake and the authors intended to cite all the included papers?

It is suggested that the excluded papers in the full-text screening phase will be cited as well.

It is suggested that interventions be mentioned in tables 2 and 3 as well.

Please cite or mention the names and authors of the related papers in Table 4.

Discussion

After including any possible papers this section needs to be revised. The categorization of this section seems to be fine, but it is suggested that authors focus more on synthesizing the results and providing a comprehensive interpretation of them. Some paragraphs are the repetition of the results and lengthy and can be better summarized.

Reviewer #2: Thank you for the opportunity to review your manuscript and I commend you for submitting it. The issue of poor oral health among people living with SMI is an important one.

Your introduction section establishes the importance of the topic, but some of the concepts and dental terminology are not clearly explained. Your study lacked a clear research question, and the inclusion criteria were very broad, so it was unsurprising that there was too much heterogeneity between studies to conduct a meta-analysis (synthesis would require more commonality between studies). Despite such broad criteria, only seven studies were included in the analysis, and two of those were published in the 1970s. The results were presented more like a simple descriptive summary of studies, which minimises the usefulness of the findings. Unfortunately, I do not think this study contributes valuable new findings for practice or research and am therefore not recommending it be accepted for publication.

In considering the quality of the writing, I found myself confused on occasion as some of the comments were rather vague or unclear, and the narrative of the manuscript did not flow well. There were some structural issues with paragraphs and sentences. Try to avoid anthropomorphisms e.g. ‘three studies had some concerns…’ These problems with structure and writing conventions can be easily rectified.

Reviewer #3: Oral health improvement is important for persons with severe mental illness. This review article is well structured and the manuscript content is detailed. Furthermore, it provides more insight for practice and can facilitate further research to enhance oral hygiene for this disadvantaged patient group. However, several points must be addressed before this manuscript can be published.

1. The authors reviewed relevant articles that are published prior to May 2020. I think that the following two articles are relevant and recommend that they be included in the literature review:

(1) Mun, S.J. et al. Reduction in dental plaque in patients with mental disorders through the dental hygiene care programme. Int J Dent Hygiene 2013;12(2):133–40.

(2) Yoshii H, et al. Effects of an educational intervention on oral hygiene and self-care among people with mental illness in Japan: a longitudinal study. BMC Oral Health 2017; 17:81.

Please consider adding these two articles into the literature review.

2. Moreover, if the authors had extended the scope of the literature review to the end of 2020, the review could have included an additional cluster randomized controlled trial (RCT) published by Kuo et al. (Effectiveness of oral health promotion program for persons with severe mental illness: a cluster randomized controlled study. BMC Oral Health 2020; 20:290).

I believe that if the authors had included the abovementioned three articles, the manuscript would be more comprehensive.

3. In the 2nd paragraph of the discussion section, the authors mentioned that oral health education might increase individual awareness of personal oral health in a negative manner. However, Teng, P.R. et al. (Gen Hosp Psychiatry 2011; 33: 253-259) showed that the neglect of oral health in persons with severe mental illness is in part due to the lack of awareness. Specifically, nearly 40% of the psychiatric inpatients rated their oral health as good despite the presence of periodontal pockets. Therefore, a dissatisfactory perception of oral health in an intervention group might not be negative because it could enhance their awareness of poor oral health and motivate them to take actions to improve oral hygiene.

**Do you want your identity to be public for this peer review?**

Reviewer #1: No

Reviewer #2: No

Reviewer #3: **Yes: **Po-Ren Teng

---

## [Author Response · Author response to Decision Letter 0]

27 Aug 2021

Dear Dr Lalloo,

Thank you for considering our study for publication in PLOS ONE and many thanks to the reviewers for their constructive feedback, which has been incredibly valuable in improving the manuscript. In response to the feedback, we have summarised the comments and our responses below.

Please ensure that your manuscript meets PLOS ONE's style requirements, including those for file naming 

The title, headings, tables and file names have been amended to ensure that the style requirements have been met.

We note that you have indicated that data from this study are available upon request. 

As all key data is included in the manuscript, this section has now been amended to state “All relevant data are within the manuscript and its Supporting Information files” 

Reviewer One

It is suggested that the participant, intervention, comparison, and outcome (PICO) framework be used to formulate the research focused question. In that way, readers will have a more clear idea of what to expect from the paper. 

This has now been clarified in this format (pages 6 and 7)

Page5, L 88. What authors mean by adults? What is the age range? 

This has now been clarified to state adults are defined as aged over 18 (page 6, L 99 – clean copy of manuscript)

Page 5, L 90. It is suggested that authors describe more in detail the other psychotic disorder since readers in the field of dentistry may not be familiar with them. 

Thank you for this insight. As the term “psychotic disorders” encompasses many different potential diagnoses, these have not all been listed, but the term has been clarified and an example provided (page 6, L 101-102).

Authors need to clearly define inclusion and exclusion criteria under separate subheadings so it would be easier for readers to follow. 

The inclusion and exclusion criteria have now been separated for clarity (page 7, L 119-122)

Page 5 L 92. What do authors mean by dual diagnosis exactly? SMI and another psychological disorder? 

This has been clarified to now state “dual psychiatric diagnosis” (page 6, L 105)

Authors need to mention the exact month for their systemic search in 2020 for each database. 

This has now been included (page 7 L 130).

Why 90% of the screening of titles and abstracts were done by one author? It’s not a common approach in conducting systematic reviews and usually 2 authors screen all the records. Moreover, does 100% agreement between the two authors, when piloting for 10% of the records, refer to those papers that were excluded, included, or both? If it only refers to excluded papers, it can be a source of bias. 

The search has been repeated with 100% of title and abstract screening undertaken independently by two reviewers to reduce the risk of bias (page 8, L135-137)

Only seven papers were included in this review which consequently prevents authors from conducting meta-analysis due to heterogeneity. By a quick search only in PubMed, the following papers were found that possibly could have been included in this review. 

The search has now been repeated, meaning that the review now has 12 studies in total. Thank you for taking the time to check for eligible studies, however some of the suggested studies did not meet the study criteria (e.g. study protocol, qualitative study), however one of the suggested studies that had been published after the initial search has now been included in the review. 

Page 8 L 162. Only one reference in the bracket was mentioned which seems to be by mistake and the authors intended to cite all the included papers? 

Thank you – citations of all the studies have now been included, and the additional citation at the end of this section is to reference the PRISMA flowchart (page 10 L 182 and 185).

It is suggested that the excluded papers in the full-text screening phase will be cited as well. 

These have now been included (page 10 L 183-185)

It is suggested that interventions be mentioned in tables 2 and 3 as well. 

These have now been included

Please cite or mention the names and authors of the related papers in Table 4. 

Citations have now been incorporated into Table 4

Discussion: After including any possible papers this section needs to be revised. The categorization of this section seems to be fine, but it is suggested that authors focus more on synthesizing the results and providing a comprehensive interpretation of them. Some paragraphs are the repetition of the results and lengthy and can be better summarized. Thank you for the helpful feedback. In addition to incorporating the results from the five additional studies, the elements of the discussion that are more repetitive (particularly in relation to the results of studies) have been removed. 

Reviewer Two

Your introduction section establishes the importance of the topic, but some of the concepts and dental terminology are not clearly explained. 

Some of the clinical terms and concepts in the introduction have been explained or have been replaced with clearer language. 

Your study lacked a clear research question, and the inclusion criteria were very broad, so it was unsurprising that there was too much heterogeneity between studies to conduct a meta-analysis (synthesis would require more commonality between studies). 

The research question has been clarified within the introduction (page 5, L 87-89) and through the inclusion criteria presented using the PICO framework (pages 6 and 7). We appreciate that the inclusion criteria were very broad, however we have tried to clarify the rationale for this within the introduction. 

Despite such broad criteria, only seven studies were included in the analysis, and two of those were published in the 1970s. The results were presented more like a simple descriptive summary of studies, which minimises the usefulness of the findings. Unfortunately, I do not think this study contributes valuable new findings for practice or research and am therefore not recommending it be accepted for publication.

The authors have repeated the search for the review, in the light of more recent studies contributing to the findings of this review. Although we acknowledge that it would have been preferable to have a meta-analysis if possible, it was felt that it would have still been inappropriate to do so, even with the increased number of studies due to the amount of heterogeneity. However, there are now some stronger aspects of the evidence base provided by the additional studies and do hope that this review can provide a summary of the evidence around oral health interventions in this population.

In considering the quality of the writing, I found myself confused on occasion as some of the comments were rather vague or unclear, and the narrative of the manuscript did not flow well. There were some structural issues with paragraphs and sentences. Try to avoid anthropomorphisms e.g. ‘three studies had some concerns…’ These problems with structure and writing conventions can be easily rectified. 

Thank you for the constructive feedback. Some of the narrative has now been amended to provide more clarity and flow, and reduce repetition (for example within the introduction and discussion). The risk of bias assessments have now been simplified to avoid unclear narrative and anthropomorphisms, thank you for noting this.

Reviewer Three

The authors reviewed relevant articles that are published prior to May 2020. I think that the following two articles are relevant and recommend that they be included in the literature review: 

Thank you very much for the helpful advice. The article by Mun et al 2014 has been included in this revision of the review, however unfortunately the article by Yoshii did not meet the inclusion criteria, so was excluded on full text screening.

Moreover, if the authors had extended the scope of the literature review to the end of 2020, the review could have included an additional cluster randomized controlled trial (RCT) published by Kuo et al. (Effectiveness of oral health promotion program for persons with severe mental illness: a cluster randomized controlled study. BMC Oral Health 2020; 20:290). I believe that if the authors had included the abovementioned three articles, the manuscript would be more comprehensive. 

Thank you very much for this suggestion. The search has now been updated, so this article has been included in the revision of this review, in addition to four other articles (including Mun et al).

In the 2nd paragraph of the discussion section, the authors mentioned that oral health education might increase individual awareness of personal oral health in a negative manner. However, Teng, P.R. et al. (Gen Hosp Psychiatry 2011; 33: 253-259) showed that the neglect of oral health in persons with severe mental illness is in part due to the lack of awareness. Specifically, nearly 40% of the psychiatric inpatients rated their oral health as good despite the presence of periodontal pockets. Therefore, a dissatisfactory perception of oral health in an intervention group might not be negative because it could enhance their awareness of poor oral health and motivate them to take actions to improve oral hygiene. 

Thank you – this is a very valid point and demonstration of very useful findings. These research findings have now been incorporated into the discussion with reference (page 29-30, L 448-450 on clean version of manuscript)

If you have any further queries, please do not hesitate to get in touch.

Yours Sincerely

Alex Macnamara

---

## [Decision Letter · Decision Letter 1]

13 Sep 2021

PONE-D-21-11259R1Improving oral health in people with severe mental illness (SMI): a systematic reviewPLOS ONE

Dear Dr. Macnamara,

Thank you for submitting your manuscript to PLOS ONE. After careful consideration, we feel that it has merit but does not fully meet PLOS ONE’s publication criteria as it currently stands. Therefore, we invite you to submit a revised version of the manuscript that addresses the points raised during the review process. A reviewer is concerned about the potential of research papers missing in the review; and recommends this is carefully reviewed. The submission can be further improved, i.e. language, and flow of information. Please submit your revised manuscript by Oct 28 2021 11:59PM. If you will need more time than this to complete your revisions, please reply to this message or contact the journal office at plosone@plos.org. Please include the following items when submitting your revised manuscript:A rebuttal letter that responds to each point raised by the academic editor and reviewer(s). You should upload this letter as a separate file labeled 'Response to Reviewers'.A marked-up copy of your manuscript that highlights changes made to the original version. You should upload this as a separate file labeled 'Revised Manuscript with Track Changes'.An unmarked version of your revised paper without tracked changes. You should upload this as a separate file labeled 'Manuscript'.If applicable, we recommend that you deposit your laboratory protocols in protocols.io to enhance the reproducibility of your results. Protocols.io assigns your protocol its own identifier (DOI) so that it can be cited independently in the future. For instructions see: https://journals.plos.org/plosone/s/submission-guidelines#loc-laboratory-protocols. Additionally, PLOS ONE offers an option for publishing peer-reviewed Lab Protocol articles, which describe protocols hosted on protocols.io. Read more information on sharing protocols at https://plos.org/protocols?utm_medium=editorial-email&utm_source=authorletters&utm_campaign=protocols.

We look forward to receiving your revised manuscript.

Kind regards,

Ratilal Lalloo

Academic Editor

PLOS ONE

Journal Requirements:

Reviewers' comments:

1. Previous round of review comments addressed? 

Reviewer #1: (No Response)

Reviewer #3: All comments have been addressed

2. Is the manuscript technically sound, and do the data support the conclusions?

Reviewer #1: Partly

Reviewer #3: Yes

3. Has the statistical analysis been performed appropriately and rigorously? 

Reviewer #1: N/A

Reviewer #3: N/A

4. Have the authors made all data underlying the findings in their manuscript fully available?

Reviewer #1: Yes

Reviewer #3: Yes

5. Is the manuscript presented in an intelligible fashion and written in standard English?

Reviewer #1: Yes

Reviewer #3: Yes

6. Review Comments to the Author

Reviewer #1: Thank you very much for submitting the revision of the paper entitled “Improving oral health in people with severe mental illness (SMI): a systematic review.” I can see that the manuscript has been improved compared to the previous version.

When reporting the results using PICO make sure to mention in advance in the text in the Method section and used the appropriate reference. Also, the main research question should be formulated before going into detail about categories. Please see similar papers in the literature to get familiar with the proper reporting format.

Please use the subheading “Inclusion Criteria” instead of “Study Design and Language”

In the response letter, the authors mentioned that qualitative studies were not considered eligible. If this is the case, please mentioned it in your exclusion criteria.

Based on the Prisma flowchart, out of the 5,330 screened records, only 16 studies were considered eligible for full-text reading which is proportionally a low rate implying a strict screening process. As a suggestion, please go through those papers you find closely related and evaluate their eligibility based on the full-text reading to make sure no study is missing. If the number of excluded is fairly small (10 or under), please use a table or explain in more detail the reason for exclusion for each paper specifically.

Regarding the quality of writing and academic style, some editing is needed since the flow is lost in some sections.

Reviewer #3: In the section of abstract, "There is a currently a lack of evidence to suggest..." The double "a" should be deleted to one "a".

7. **Do you want your identity to be public for this peer review?**

Reviewer #1: No

Reviewer #3: No

---

## [Author Response · Author response to Decision Letter 1]

17 Oct 2021

Comment: Please review your reference list to ensure that it is complete and correct 

Response:

Reference 1 URL updated. 

Reference 7 has been updated to include the URL. 

Reference 10 URL updated.

Duplicate reference noted (5 and 11) – duplicate removed – reference 5 now only reference to this site. 

Reference 17 updated to correct author names.

Reference 18 URL updated.

Reference 23 URL updated.

Reference 36 URL updated.

Reference 42 URL updated.

Reference 44 URL updated.

Reference 48 URL updated.

Reference 85 URL updated.

Reviewer One

Comment: When reporting the results using PICO make sure to mention in advance in the text in the Method section and used the appropriate reference. Also, the main research question should be formulated before going into detail about categories. Please see similar papers in the literature to get familiar with the proper reporting format. 

Response: Thank you for this helpful suggestion. Following the research aim at the end of the introduction (page 5, lines 88-90 – clean copy), the methods section has now been updated to reflect the use of the PICO framework with the relevant reference (page 6, lines 98-100).

Comment:Please use the subheading “Inclusion Criteria” instead of “Study Design and Language” 

Response:This has now been changed as suggested (Page 7, line 117)

Comment: In the response letter, the authors mentioned that qualitative studies were not considered eligible. If this is the case, please mentioned it in your exclusion criteria. 

Response: This has now been included for clarity, thank you. Please see page 7 line 125.

Comment: Based on the Prisma flowchart, out of the 5,330 screened records, only 16 studies were considered eligible for full-text reading which is proportionally a low rate implying a strict screening process. As a suggestion, please go through those papers you find closely related and evaluate their eligibility based on the full-text reading to make sure no study is missing. If the number of excluded is fairly small (10 or under), please use a table or explain in more detail the reason for exclusion for each paper specifically. 

Response: Thank you for noting this. 

The 5,330 records encompassed both duplicate records and conference abstracts, meaning that in total 1,465 records from the database search were screened, with 16 identified for full text screening. The search was also broad as it aimed to encompass the wide range of potential interventions and outcomes, therefore we were anticipating that there would be a number of unrelated articles.

The reasons for exclusion from the full text screening are listed with references (page 10, lines 186-189).

Comment: Regarding the quality of writing and academic style, some editing is needed since the flow is lost in some sections. 

Response: Changes have been made to the writing in order to improve the flow and ensure clarity and consistency.

Reviewer #3: 

Comment: In the section of abstract, "There is a currently a lack of evidence to suggest..." The double "a" should be deleted to one "a".

Response: Thank you for making us aware of this, this has now been amended to read "There is currently a lack of evidence to suggest..." (Page 2 – line 23)

---

## [Decision Letter · Decision Letter 2]

1 Nov 2021

PONE-D-21-11259R2Improving oral health in people with severe mental illness (SMI): a systematic reviewPLOS ONE

Dear Dr. Macnamara,

Thank you for submitting your manuscript to PLOS ONE. After careful consideration, we feel that it has merit but does not fully meet PLOS ONE’s publication criteria as it currently stands. Therefore, we invite you to submit a revised version of the manuscript that addresses the points raised during the review process. Please submit your revised manuscript by Dec 16 2021 11:59PM. If you will need more time than this to complete your revisions, please reply to this message or contact the journal office at plosone@plos.org. Please include the following items when submitting your revised manuscript:A rebuttal letter that responds to each point raised by the academic editor and reviewer(s). You should upload this letter as a separate file labeled 'Response to Reviewers'.A marked-up copy of your manuscript that highlights changes made to the original version. You should upload this as a separate file labeled 'Revised Manuscript with Track Changes'.An unmarked version of your revised paper without tracked changes. You should upload this as a separate file labeled 'Manuscript'.If applicable, we recommend that you deposit your laboratory protocols in protocols.io to enhance the reproducibility of your results. Protocols.io assigns your protocol its own identifier (DOI) so that it can be cited independently in the future. For instructions see: https://journals.plos.org/plosone/s/submission-guidelines#loc-laboratory-protocols. Additionally, PLOS ONE offers an option for publishing peer-reviewed Lab Protocol articles, which describe protocols hosted on protocols.io. Read more information on sharing protocols at https://plos.org/protocols?utm_medium=editorial-email&utm_source=authorletters&utm_campaign=protocols.

We look forward to receiving your revised manuscript.

Kind regards,

Ratilal Lalloo

Academic Editor

PLOS ONE

Journal Requirements:

Reviewers' comments:

Reviewer's Responses to Questions

**Comments to the Author**

1. Adequately addressed reviewers' comments?

Reviewer #1: (No Response)

Reviewer #3: All comments have been addressed

2. Is the manuscript technically sound, and do the data support the conclusions?

Reviewer #1: Partly

Reviewer #3: Yes

3. Has the statistical analysis been performed appropriately and rigorously? 

Reviewer #1: N/A

Reviewer #3: N/A

4. Have the authors made all data underlying the findings in their manuscript fully available?

Reviewer #1: Yes

Reviewer #3: Yes

5. Is the manuscript presented in an intelligible fashion and written in standard English?

Reviewer #1: Yes

Reviewer #3: Yes

6. Review Comments to the Author

Reviewer #1: Thank you very much for submitting the revison of the manuscript.

Some of the comments have been address in this versin, but authors need to excersise more caution when reporting the results and findings so that the results will be homogeneous throughout the manuscript. As mentioned before, the Prisma flowchart has not been revised. It clearly mentions in your Prisma flowchart that 5,333 records have been obtained after the dublicates have been removed, and the whole screening process is based on the 5,330 record!

Also in the reposnse letter authors mentioned 1,465 unique record were identified (if this is the case, this number should be reported in the abstract as well). However in the manuscript this number is 1,462! and again 2,073 was mentioned instead of 5,330 in the text for numbers of records before excluding the duplicates! In my opinion, the authors need to address these controversies since for potential readers it will be not possible to know what approch the authors took to select the final inluded studies.

Reviewer #3: The search has now been updated, more articles have been included in the revised manuscript of this review, therefore, I have no more comments.

7. Reviewer #1: No

Reviewer #3: No

---

## [Author Response · Author response to Decision Letter 2]

11 Nov 2021

Reviewer comment: “Some of the comments have been address in this versin, but authors need to excersise more caution when reporting the results and findings so that the results will be homogeneous throughout the manuscript. As mentioned before, the Prisma flowchart has not been revised. It clearly mentions in your Prisma flowchart that 5,333 records have been obtained after the dublicates have been removed, and the whole screening process is based on the 5,330 record!

Also in the reposnse letter authors mentioned 1,465 unique record were identified (if this is the case, this number should be reported in the abstract as well). However in the manuscript this number is 1,462! and again 2,073 was mentioned instead of 5,330 in the text for numbers of records before excluding the duplicates! In my opinion, the authors need to address these controversies since for potential readers it will be not possible to know what approch the authors took to select the final inluded studies.”

Response: Thank you very much for your feedback on this reporting of the search strategy

To avoid confusion, we have amended the flow chart so that it only includes records identified from the database and grey literature search. We now only describe the hand searching of conference proceedings in the text (as this did not identify any records). We hope this much more clearly reports the screening process.

---

## [Decision Letter · Decision Letter 3]

17 Nov 2021

Improving oral health in people with severe mental illness (SMI): a systematic review

PONE-D-21-11259R3

Dear Dr. Macnamara,

We’re pleased to inform you that your manuscript has been judged scientifically suitable for publication and will be formally accepted for publication once it meets all outstanding technical requirements.

Kind regards,

Ratilal Lalloo

Academic Editor

PLOS ONE

**Comments to the Author**

1. If the authors have adequately addressed your comments.

Reviewer #1: (No Response)

Reviewer #3: All comments have been addressed

2. Is the manuscript technically sound, and do the data support the conclusions?

Reviewer #1: Yes

Reviewer #3: Yes

3. Has the statistical analysis been performed appropriately and rigorously? 

Reviewer #1: N/A

Reviewer #3: N/A

4. Have the authors made all data underlying the findings in their manuscript fully available?

Reviewer #1: Yes

Reviewer #3: Yes

5. Is the manuscript presented in an intelligible fashion and written in standard English?

Reviewer #1: Yes

Reviewer #3: Yes

6. Review Comments to the Author

Reviewer #1: (No Response)

Reviewer #3: The authors had answered the comments from the reviewers and the study had included all available papers related to study topic. I have no further comment.

7. Reviewer #1: No

Reviewer #3: No

---

## [Editor Report · Acceptance letter]

19 Nov 2021

PONE-D-21-11259R3 

Improving oral health in people with severe mental illness (SMI): a systematic review 

Dear Dr. Macnamara:

I'm pleased to inform you that your manuscript has been deemed suitable for publication in PLOS ONE. Congratulations! Your manuscript is now with our production department. 

Kind regards, 

on behalf of

Dr. Ratilal Lalloo 

Academic Editor

PLOS ONE